# Triple ionization and fragmentation of benzene trimers following ultrafast intermolecular Coulombic decay

Jiaqi Zhou [1], Xitao Yu[2], Sizuo Luo [2], Xiaorui Xue [1], Shaokui Jia[1], Xinyu Zhang[2], Yongtao Zhao [1], Xintai Hao [1], Lanhai He [2], Chuncheng Wang [2], Dajun Ding [2] ✉ & Xueguang Ren [1] ✉

Intermolecular interactions involving aromatic rings are ubiquitous in biochemistry and they govern the properties of many organic materials. Nevertheless, our understanding of the structures and dynamics of aromatic clusters remains incomplete, in particular for systems beyond the dimers, despite their high presence in many macromolecular systems such as DNA and proteins. Here, we study the fragmentation dynamics of benzene trimer that represents a prototype of higher-order aromatic clusters. The trimers are initially ionized by electron-collision with the creation of a deep-lying carbon $2s^{-1}$ state or one outer-valence and one inner-valence vacancies at two separate molecules. The system can thus relax via ultrafast intermolecular decay mechanisms, leading to the formation of $C_6H_6^+ \cdot C_6H_6^+ \cdot C_6H_6^+$ trications and followed by a concerted three-body Coulomb explosion. Triple-coincidence ion momentum spectroscopy, accompanied by ab-initio calculations and further supported by strong-field laser experiments, allows us to elucidate the details on the fragmentation dynamics of benzene trimers.

Aromatic rings are prevalent throughout nature and play a vital role in many areas of chemistry, biology, and material science[1-5]. This is particularly relevant for the aromatic $\pi$–$\pi$ interactions, which can contribute substantially to protein folding, DNA base stacking, self-assembly, drug binding, crystal engineering, and so on[6-9]. In this respect, benzene (Bz, $C_6H_6$) dimers have been studied extensively for understanding the fundamental physics of individual $\pi$–$\pi$ interactions[10-16]. In contrast, because of the increased difficulty in the isolation of artificial model systems, investigations on the structures and dynamics of higher-order aromatic clusters are very incomplete and largely limited to theoretical studies[17-24].

In biological systems, aromatic rings are often involved in more than one $\pi$–$\pi$ interaction at a time, such as the stacking of nucleic acid bases in the double-helical structure of DNA[21]. In proteins as well, aromatic trimers are highly relevant for protein stability and molecular recognition processes due to the presence of aromatic trimer motifs in

half of all proteins, and such motifs are built by adopting the same structures found for benzene trimers in a vacuum, which is thought of as the basic building unit of higher-order aromatic clusters[6,22,23]. It is therefore critical to elucidate the properties and dynamics of benzene trimers, which are of interest to a wide variety of research fields ranging from macromolecular sciences to astrochemistry[1-5,25,26].

During the past two decades, there has been intense research on the properties of excited states in various clusters of atoms and molecules. An important motivation is the potential for opening ultrafast decay mechanisms involving the neighbors in close vicinity. For ionization in an inner-valence shell, intermolecular Coulombic decay (ICD), which is a nonlocal autoionization process first predicted in ref. 27, may become operative. In ICD, the inner-valence vacancy is filled by an electron from an outer shell and the de-excitation energy is transferred to a neighboring unit causing further ionization, see e.g., refs. 28–30. This decay mechanism is fast and evolves on a

[1]MOE Key Laboratory for Nonequilibrium Synthesis and Modulation of Condensed Matter, School of Physics, Xi'an Jiaotong University, Xi'an 710049, China.
[2]Institute of Atomic and Molecular Physics, Jilin University, Changchun 130012, China. ✉e-mail: dajund@jlu.edu.cn; renxueguang@xjtu.edu.cn

femtosecond (fs) timescale[31–33]. It leads to the ejection of a low-energy secondary electron and the formation of two repulsive ions at a distance of few Ångströms (Coulomb explosion), which can play an important role in radiation biology and chemistry[34–37].

In the present work, we study the electron-collision (260 eV) induced ionization process in benzene trimer $(C_6H_6)_3$ and the subsequent reaction dynamics of triply-charged ions. Here, we used electrons as bombarding particles because of their essential role in the radiation effects in gases and condensed matter[37,38]. The present studies through fragment ions coincident momentum spectroscopy, supported by strong-field fs laser experiments and ab-initio molecular dynamics simulations in which we consider eight different conformational geometries of the neutral benzene trimer[18], provide the mechanistic details about the fragmentation dynamics of benzene trimers.

## Results

### Triple ionization mechanisms of $(C_6H_6)_3$

In our experiments, $C_6H_6^+ \cdot C_6H_6^+ \cdot C_6H_6^+$ trications can potentially be produced via four channels, i.e., (i) The sequential ionization (SI), where the projectile electron successively kicks out one outer-valence electron from each molecule of the trimer, and thus leads to the repulsive tricationic state (shown in Fig. 1a). The minimum energy required for SI is estimated as the sum of the single-ionization potential of three molecules (9.2 eV × 3)[39] plus the Coulomb energy (about 8.4 eV). This gives roughly 36.0 eV for the threshold energy of $C_6H_6^+ \cdot C_6H_6^+ \cdot C_6H_6^+$ trication, which is in line with our calculation of the triple-ionization potential of benzene trimer (35.9 eV) at the CCSD(T)/cc-pVDZ level (see Methods); (ii) The double sequential ionization plus ICD (dSI + ICD), i.e., one outer-valence and one inner-valence vacancies are created separately at two molecules of the trimer and the following ICD process causes the ionization of the third molecule (shown in Fig. 1b). In dSI + ICD, an appropriate inner-valence vacancy, i.e., $2e_{1u}^{-1}$ state with binding energy around 24.0 eV[40], is required to make the following ICD channel accessible.

The channel (iii) is the so-called double ICD (dICD) process[41–43]. As illustrated in Fig. 1c, the dICD reaction in benzene trimer is initiated by electron-collision with the removal of a deep-lying carbon 2s (C2s) inner-valence electron with binding energy above 36.0 eV. Afterward, the $C_6H_6^{+*}(C2s^{-1}) \cdot (C_6H_6)_2$ ionic state undergoes the dICD process, i.e., an electron from the outer-valence shell of $C_6H_6^{+*}$ fills the $C2s^{-1}$ vacancy, and the energy released (>26.8 eV) is sufficient to each single ionize the neighboring two molecules. Three $C_6H_6^+$ cations that are formed in the decay process repel each other, leading to the Coulomb explosion of the system. We notice that the energy released may also singly or doubly ionize one neighboring molecule, resulting in $C_6H_6^+ \cdot C_6H_6^+ \cdot C_6H_6$ and $C_6H_6^+ \cdot C_6H_6^{++} \cdot C_6H_6$ states, respectively. However, neither of them can contribute to the present results, which demands three $C_6H_6^+$ ions to be detected in coincidence, and the latter process requires higher threshold energy of about 39.4 eV to be accessible; (iv) Electron transfer mediated decay (ETMD)[44–48] in which the outer-valence and inner-valence electrons are stripped from one molecule of the trimer. An electron from the neighboring molecules is transferred to fill the initial inner-valence vacancy and the energy released causes the ionization of the other neutral benzene (see in Fig. 1d). There are three candidate states $2e_{1u}^{-1}1e_{1g}^{-1}$, $2e_{1u}^{-1}3e_{2g}^{-1}$, and $2a_{1g}^{-1}1a_{2u}^{-1}$ lying above 36.0 eV binding energy[49], which are energetically accessible for the ETMD channel. It should be noted that since the decay width decreases exponentially with the intermolecular distance (R), the larger distance of R ~5.15 Å in benzene trimer can lead to a lower ETMD efficiency in comparison with that in the noble gas clusters[44–46]. Moreover, the molecule in these dicationic states may relax undergoing ultrafast Coulomb explosion prior to the electron transfer, in particular for the processes involving proton emission[50], which can suppress the occurrence of ETMD.

Our analysis of the calculated absolute cross-sections indicates the SI is only a minor channel (~5.9 × 10⁻⁴ Å²) in comparison with the dSI+ICD (~7.4 × 10⁻³ Å²) and dICD (~1.28 × 10⁻² Å²) processes (see Methods), while the contribution from the ETMD channel might be small mainly because of the competing Coulomb explosion processes of the molecule in the initial dicationic states mentioned above.

### Three-body dissociation pathway

The complete three-body Coulomb explosion processes of benzene trimers were measured using a multi-particle coincidence momentum spectrometer[51,52]. All three intact $C_6H_6^+$ cations are detected in coincidence. From the time-of-flight (TOF) and the position of the particle

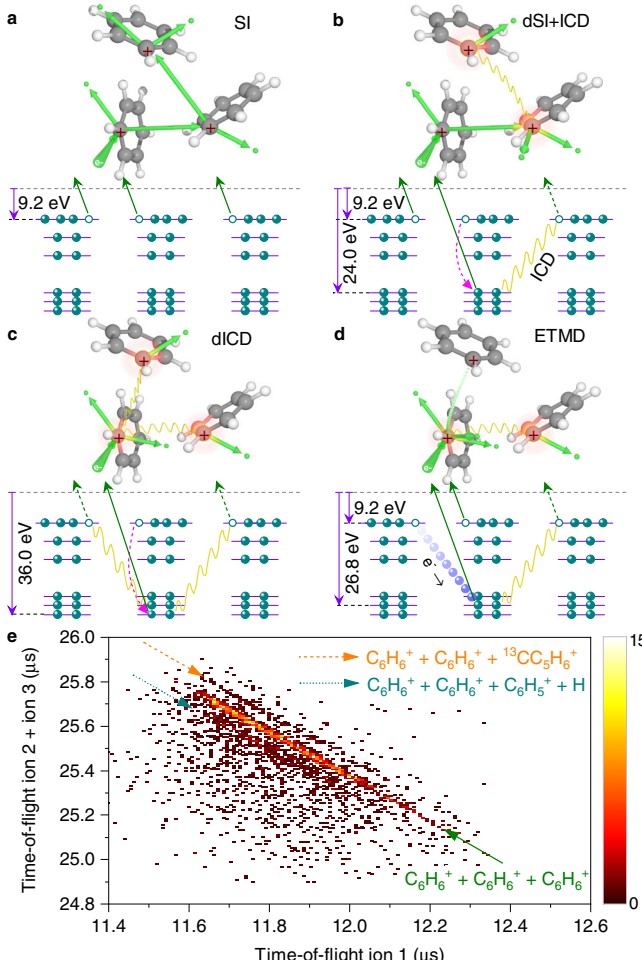

**Fig. 1 | Schematic of triple-ionization channels in benzene trimer leading to three $C_6H_6^+$ cations and measured ion time-of-flight (TOF) correlation spectrum. a** The sequential ionization (SI) of three molecules upon electron-impact; **b** The double sequential ionization creates outer-valence and inner-valence vacancies at two molecules followed by an additional intermolecular Coulombic decay (dSI + ICD) process; **c** One deep-lying inner-valence electron is ionized followed by the double ICD (dICD) process leading to the ionization of two neighboring molecules; **d** The electron transfer mediated decay (ETMD) process, where both inner-valence and outer-valence vacancies are created at one molecule, intermolecular electron transfer followed by ICD causes ionization of the third molecule. Green, blue and pink arrows denote electron emission, energy level, and intramolecular de-excitation, respectively; Yellow wavy lines: intermolecular energy transfer; Atomic colors: carbon (gray), hydrogen (white). **e** Measured TOF correlation map between the first and the sum of the second and third detected cations. The green, indigo, and orange arrows denote the complete $C_6H_6^+ + C_6H_6^+ + C_6H_6^+$ Coulomb explosion, one hydrogen loss, and the ¹³C isotope channels, respectively. The count intensity is color-coded on a linear scale. Source data in Fig. 1e are provided as a Source Data file.

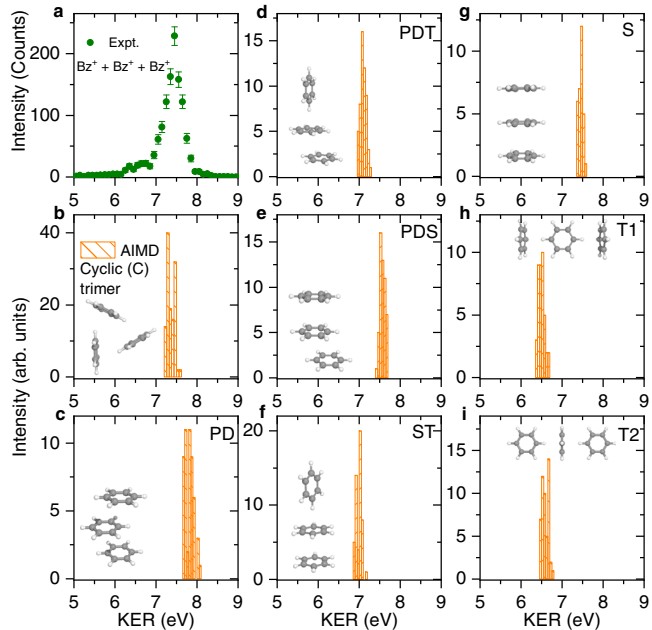

**Fig. 2 | Measured and AIMD calculated KER for the three-body Coulomb explosion of $(C_6H_6)_3^{3+}$. a** Experiment. **b–i** AIMD calculated KER for eight different conformers of the benzene trimer: C-trimer (**b**), PD (**c**), PDT (**d**), PDS (**e**), ST (**f**), S (**g**), T1 (**h**), and T2 (**i**) conformers. Error bars represent statistical s.d. and were calculated by the square root of the true coincidence counts. Source data are provided as a Source Data file.

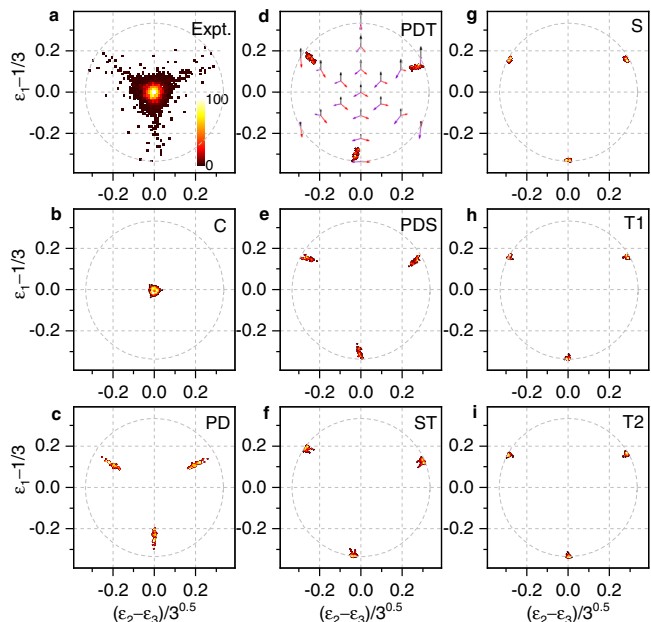

**Fig. 3 | Measured and AIMD calculated Dalitz plots. a**, Experiment. **b–i** Calculations for eight different conformers of the benzene trimer: C-trimer (**b**), PD (**c**), PDT (**d**), PDS (**e**), ST (**f**), S (**g**), T1 (**h**), and T2 (**i**) conformers. Also included in (**d**) is the calculated three-body Dalitz plot. Note that $\epsilon_j$ denotes randomly the relative energy of the three $C_6H_6^+$ ions since they are indistinguishable. The color bar is linear with arbitrary units. Source data are provided as a Source Data file.

hitting on the detector, the initial three-dimensional momentum vector (which is equivalent to the kinetic energies and emission directions) of each detected particle is reconstructed (see Methods). Due to momentum conservation, the momentum sum of all three $C_6H_6^+$ cations is close to zero, with a narrow distribution determined mainly by the recoil momentum of the impinging electron during the collision. We identify the three-body Coulomb explosion channel using a TOF correlation map between the first and the sum of the second and third detected cations, which is shown in Fig. 1e. As the TOF of an ion depends on its momentum and mass, the three-body Coulomb explosion channel ($C_6H_6^+ + C_6H_6^+ + C_6H_6^+$) leads to a sharp diagonal line in the TOF correlation map. In addition, we observe also the fragmentation channels due to one hydrogen loss and the $^{13}C$ isotope, which are well separated from the $C_6H_6^+ + C_6H_6^+ + C_6H_6^+$ fragmentation channel studied here.

## Kinetic energy release spectra

The measured kinetic energy release (KER) of the three-body Coulomb explosion channel is presented in Fig. 2a, which shows a dominant peak at about 7.4 eV and a small shoulder at around 6.5 eV. As the intermolecular potential drops with 1/R assuming two point-like charges separated by the distance R, we may deduce the Coulomb energy by considering that the two charges are located at the center-of-mass (COM) of the molecules. For a symmetric triangular or cyclic (C) structure of the trimer, which is suggested to be the lowest-energy configuration[17–19], we determine the intermolecular COM distance of about 5.15 Å. This would result in the total KER of around 8.4 eV (3 × 1/R) as each benzene molecule in the trimer is in direct interaction with the other two benzene molecules. This value lies about 1.0 eV higher than the measured KER, indicating that some amount of the Coulomb repulsion energy might be transferred to the rotational and vibrational motions of the molecular ions[53,54].

To quantify this process, we have performed ab-initio molecular dynamics (AIMD) simulations starting from the sampled trimers with a given initial temperature (30 K) and instantaneous removal of the

three outermost electrons from each of the molecules (see Methods). It should be noted that this approach considers the ICD process to be instantaneous and omits any intermolecular dynamics in the intermediate states. Here, we consider eight different conformers of the neutral trimer, which consist of various combinations of the prototypical configurations of the benzene dimer: the cyclic (C), the sandwich (S), T-shaped (T), and parallel-displaced (PD) configurations[11,12]. The calculated KER distributions are presented in Fig. 2b–i, which show single-peak structures centered at around 7.4, 7.8, 7.1, 7.6, 7.0, 7.5, 6.5, and 6.6 eV for the C-trimer, PD, PDT, PDS, ST, S, T1, and T2 conformers, respectively. The molecular structures for different conformers are also presented in the insets of the figure.

## Dalitz plots

For a further analysis of the three-body dissociation mechanism, we present the experimental and AIMD calculated Dalitz plots[55] in Fig. 3 and Newton diagrams[56] in Fig. 4. Here, the Dalitz plot shows a probability-density in terms of the vector correlation of three fragment ions, i.e.,: $\epsilon_1 - 1/3$ versus $(\epsilon_2 - \epsilon_3)/\sqrt{3}$, where $\epsilon_j$ with j = 1, 2, 3 denotes randomly the relative energy of three $C_6H_6^+$ fragments since they are indistinguishable. It is defined as $\epsilon_j = p_j^2/(2 \times m_j \times KER)$, i.e., the kinetic energy as a fraction of the total KER. Due to momentum conservation, the events are restricted to lie within a circle of radius 1/3. Each entry in this plot can be assigned to a geometrical configuration of the momentum vectors. This is shown schematically in Fig. 3d for the case of a three-body breakup with equal mass fragments.

The experimental spectrum, shown in Fig. 3a, reveals mainly an equilateral triangular pattern with the highest density in the center of the Dalitz plot. This suggests that the symmetric triangular breakup is the dominant dissociation mechanism of benzene trimers. Our AIMD calculated Dalitz plots are presented in Fig. 3b–i for different conformers of the trimers. As shown in Fig. 3b, the C-trimer calculation exhibits a pattern located in the center of the plot, which agrees well with the main experimental feature, suggesting the predominance of a triangular cyclic structure of the benzene trimer[17–19]. In addition, the

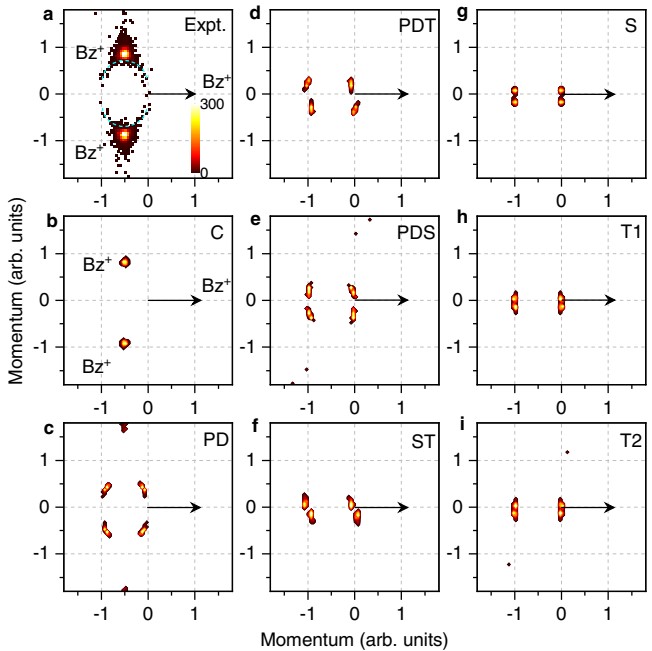

**Fig. 4 | Measured and AIMD calculated Newton diagrams. a** Experiment.
**b–i** Calculations for eight different conformers of the benzene trimer: C-trimer (**b**), PD (**c**), PDT (**d**), PDS (**e**), ST (**f**), S (**g**), T1 (**h**), and T2 (**i**) conformers. The black arrow denotes randomly the momentum of one of the three $C_6H_6^+$ fragments. The color bar is linear with arbitrary units. Source data are provided as a Source Data file.

experiment shows a wing structure beyond the main triangular pattern, which is in agreement with the simulation with a PD conformer (Fig. 3c). This result can indicate that the benzene trimers generated from a supersonic gas jet consist also a small fraction of the PD conformer.

### Newton diagrams

In Fig. 4, the three-body dissociation dynamics of benzene trimers are analyzed using the so-called Newton diagram[56], which can exhibit the momentum correlations of the three fragment $C_6H_6^+$ ions. In these diagrams, the momentum vector of one of three $C_6H_6^+$ ions is represented by an arrow fixed at one arbitrary unit in the horizontal axis. The momentum vectors of the other two $C_6H_6^+$ ions are normalized to the length of the first $C_6H_6^+$ ion momentum vector and mapped in the upper and lower half of the diagram, respectively.

The experimental data is presented in Fig. 4a, which shows mainly a symmetric triangular configuration of the three-body dissociation with additional tails marked by dashed semicircles. This spectrum reveals a concerted breakup of the triply-charged trimers[56], in which all three fragments are mutually correlated, i.e., with a relative angle of about 120°. The main experimental features are well reproduced by our AIMD simulation with a cyclic structure of the neutral trimer, which is shown in Fig. 4b. In addition, the PD calculation, shown in Fig. 4c, is in line with the observed semicircle structures (dashed curve) of the Newton diagram. While the calculations for other conformers are far from the range of the experimental pattern, in particular for the ST, S, T1, and T2 conformers, as shown in Fig. 4f–i, respectively, there may be small contributions from PDS and PDT conformers. Note that the Newton diagrams with a momentum range of ±8 (arb. units) are presented in Supplementary Fig. 2 to visualize generally the three-body dissociation dynamics, in particular for the results where the slow ion is considered as a reference. Overall, the present results of KER, Dalitz, and Newton spectra indicate that the cyclic trimer is the dominant structure of the benzene trimer with a small fraction of the PD conformer. While the contributions of the other conformers are minor in the present supersonic jet experiment.

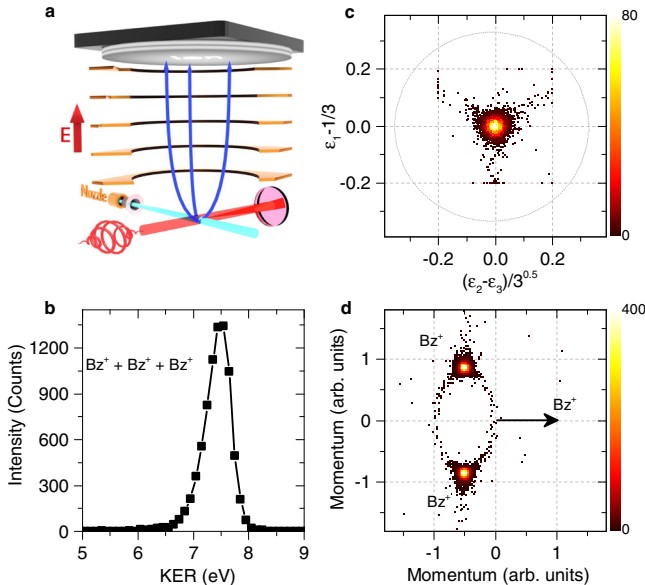

**Fig. 5 | Strong-field laser-induced fragmentation of benzene timers. a** Schematic view of the experimental geometry with a circularly polarized laser pulse (40 fs). The $(C_6H_6)_3$ trimer from supersonic expansion is triply ionized by the strong-field fs laser, leading to the formation of three $C_6H_6^+$ cations in the Coulomb explosion, which are detected in coincidence using a time- and position-sensitive detector with a uniform electric field. **b–d** Measured KER distribution (**b**), Dalitz plot (**c**), and Newton diagram (**d**) for $C_6H_6^+ + C_6H_6^+ + C_6H_6^+$ three-body Coulomb explosion channel. The color bar is linear with arbitrary units. Source data in Fig. 5b–d are provided as a source data file.

### Triple ionization of $(C_6H_6)_3$ by strong-field fs laser

To further support the present observations, we performed an additional experiment in benzene trimer using strong-field fs laser pulses[57–59], as is schematically shown in Fig. 5a. In this experiment, the repulsive $C_6H_6^+ \cdot C_6H_6^+ \cdot C_6H_6^+$ tricationic state is mainly caused by sequentially ionizing three molecules of the trimer during the laser pulse ($3 \times SI$)[60]. While the SI with additional recollision mechanisms can be ruled out as playing an important role in the triple ionization of benzene trimer since the recollision processes of the photoelectron are effectively suppressed by the circularly polarized light used in the fs laser experiment[61]. The created three $C_6H_6^+$ ions are detected in coincidence with the determinations of the momentum vector and, consequently, the kinetic energy of each fragment ions. The obtained KER, Dalitz, and Newton spectra for $C_6H_6^+ + C_6H_6^+ + C_6H_6^+$ Coulomb explosion channel are presented in Fig. 5b–d, respectively, which are in good agreement with the main features of the electron-initiated experiments. In the laser experiment, three outer-valence electrons are rapidly stripped from each molecule of the benzene trimer (<40 fs), and the nuclear dynamics during the $3 \times SI$ is nearly frozen. The subsequent Coulomb explosion of the trimer can thus lead to a KER distribution similar to the main peak obtained in the dICD and dSI + ICD processes. These results can support that the electron-collision-induced dICD and dSI + ICD processes can take place in the fs regime, which is comparable to the timescale of the $3 \times SI$ occurring in the laser experiment. It must be stressed here that the ionization mechanisms leading to the Coulomb explosion of benzene trimers are different between the electron-initiated and laser experiments, a dominant peak at KER ~7.4 eV can be observed in both experiments due to the ultrafast nature of their initial ionization processes. On the other hand, the previous observation of a small shoulder at KER ~6.5 eV (Fig. 2a) is not obtained in the fs laser experiment. In the electron-initiated experiments, a more broad distribution of the Dalitz plot is obtained for the KER region of 6.0–6.8 eV in comparison with that at the higher KER region (6.8–9 eV), which are shown in

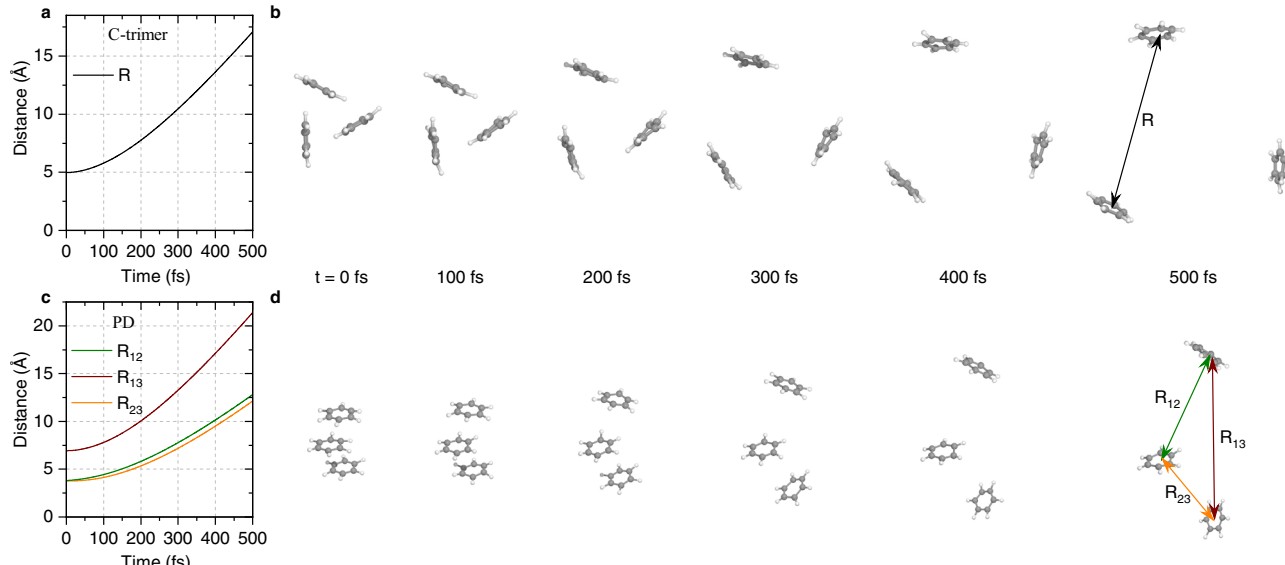

**Fig. 6 | Calculated molecular motions of the three-body Coulomb explosion processes. a, b** C-trimer calculations for the intermolecular distance (R) as a function of time (**a**) and the molecular snapshots from 0 to 500 fs (**b**). **c, d** PD conformer calculations for the intermolecular distances $R_{12}$, $R_{23}$, and $R_{13}$ as a function of time (**c**) and the molecular snapshots (**d**). Source data in Fig. 6a, c are provided as a source data file.

Supplementary Fig. 1. This observation indicates that the trimer structure might be rearranged prior to the decay mechanism in particular for the dSI+ICD channel. As in this process, after the initial dSI the two cations in benzene trimer ($C_6H_6^+ \cdot C_6H_6^{+*} \cdot C_6H_6$) begin to repel each other, which can cause an increase in the intermolecular distances before the ICD process. Since the ICD rate strongly depends on the intermolecular distance ($1/R^6$)[62], which can cause the lower yield of the shoulder at KER ~6.5 eV. According to the KER difference between the main peak (7.4 eV) and the small shoulder (6.5 eV), we can estimate a possible upper limit of the ICD lifetime in the dSI + ICD channel using a classical model, which amounts to around 239 fs (see Supplementary Fig. 3).

## Ultrafast fragmentation dynamics

The calculated fragmentation dynamics of the three-body Coulomb explosion process of benzene trimers are shown in Fig. 6. The intermolecular distances between two exploding $C_6H_6^+$ ions as a function of propagation time are presented in Fig. 6a, c for the cyclic and PD conformers of the trimers, respectively. The real-time molecular motions are presented in Fig. 6b, d, which are starting from the cyclic and PD geometries of the trimers. These spectra nicely visualize the concerted fragmentation of the $(C_6H_6)_3^{3+}$ trications in which the intermolecular distances of R ~17.0 Å for the cyclic trimer and $R_{13}$ ~21.5 Å for the PD conformer are observed at the propagation time of 500 fs. This study shows a fast and complete breakup pathway of the benzene trimers and thus offers the potential for biomolecular damage. The present observations are suggested to be considered in investigations and models on biomolecular imaging with X-rays and other radiation sources since the damage-induced sample movement can limit the resolution of structural studies on biological macromolecules[63–65].

## Discussion

The present study provides a detailed picture of the structural and dynamical properties of benzene trimer utilizing the coincident measurements of all three fragment $C_6H_6^+$ ions and ab-initio calculations. This three-body Coulomb explosion channel can be reached via different processes, depending on the initial ionization state formed by electron-collision ($E_0 = 260$ eV). Our calculations on the ionization

cross-section after electron-impact indicate that the sequential ionization (SI) of all three benzene molecules can be ruled out as a dominant mechanism for the Coulomb explosion channel while it is initiated by the creation of a deep-lying C2s hole or two holes at the outer-valence and inter-valence shells of two separate molecules. The systems can thus relax via the ultrafast dICD and dSI+ICD mechanisms, respectively, which cause further ionization of neighboring molecules and lead to the complete Coulomb explosion of the trimers. In addition, when both the outer-valence and inner-valence vacancies are located on one molecule, the ETMD process may become operative, which, however, can be suppressed mainly because of the competing Coulomb explosion of the molecule in the initial dicationic states. As a result, dSI+ICD and dICD are found to be the dominate mechanisms leading to the $C_6H_6^+ + C_6H_6^+ + C_6H_6^+$ Coulomb explosion of the trimers.

The additional strong-field fs laser experiments confirm that dSI +ICD and dICD occur on a femtosecond timescale where the ICD lifetime in the dSI+ICD channel can be estimated to be less than 239 fs. It is also to be noted that the present dSI + ICD channel (which was not considered previously) could play a more important role in the larger clusters as there are increases in the cross-section of the initial dSI process. The three-body fragmentation dynamics of benzene trimers were interpreted with the help of AIMD simulations in which we found that the calculations considering a cyclic structure of the neutral trimer reproduce well with the main features in the measured KER distribution, Dalitz plot, and Newton diagram. These results indicate that the cyclic trimer is the dominant structure of the benzene trimers, which consist also of a small fraction of the PD conformer. While the contributions of the other conformers are minor in the supersonic gas jet. Furthermore, the present study of ICD processes in benzene trimers has proven a valuable tool for imaging the structure of molecular complexes.

Our work reveals a concerted fragmentation mechanism with a dominant symmetric triangular configuration for the three-body dissociation of benzene trimers, which can be a general phenomenon occurring in biological systems. Indeed, our further calculations indicate that it may also occur in the trimers involving adenine (A) and thymine (T) bases, which form the cyclic and PD type structures of T·A·T trimers[66] (Supplementary Fig. 4). Due to the high presence of aromatic trimers with the cyclic and PD structures in proteins and

DNA[6,22,23] and the essential role of secondary electrons for the radiation effects in gases and condensed matter[37,38], the results obtained in this study could have important implications for a deeper and more complete understanding of radiation biology at the molecular level.

## Methods

### Triple-coincidence ion momentum spectroscopy

The experiment was performed using a multi-particle coincidence momentum spectrometer (reaction microscope)[51,52]. In this apparatus, the benzene trimers are generated through supersonic gas expansions of helium (1 bar) with seeded benzene vapor at room temperature (~300 K). This molecular gas jet is crossed with a pulsed electron beam (260 eV) from a photoemission electron gun (UV-light with 0.5 ns pulse duration) consisting of a tantalum cathode[67]. Three $C_6H_6^+$ cations originating from the Coulomb explosion of $(C_6H_6)_3^{3+}$ are guided by a homogeneous electric field (25 V cm⁻¹) to a time- and position-sensitive microchannel plate detector with hexagonal delay-line position read-out. From the time-of-flight (TOF) and the position of impact on the detector, the initial three-dimensional momentum vector of each detected particle (which is equivalent to the kinetic energies and emission directions) is determined. The triple-coincidence measurement is also applied in the strong-field fs laser experiments, where the circularly polarized laser pulse centered at 800 nm with a temporal duration of 40 fs was generated from an amplified Ti:sapphire laser system. The laser beam was focused on the center of the gas jet to ionize the benzene trimer and the peak density of the pulse was estimated to be I ~6 × 10¹⁴ W cm⁻² [57–59].

### Molecular dynamics simulation

Our ab-initio molecular dynamics (AIMD) simulations were performed using the Gaussian package[68]. We consider the geometries of eight different conformers of the benzene trimer[18]. First, the initial conditions, i.e., geometries and velocities of every atom, of the neutral trimer were sampled by the quasi-classical fixed normal-mode sampling method under the temperature of 30 K (supersonic gas jet) in which the populations of the initial vibrational states were determined by Boltzmann distributions. Second, molecular dynamics simulations were performed under the extended Lagrangian molecular dynamics scheme adopting the so-called atom-centered density matrix propagation (ADMP) method using the density-functional theory method at B3LYP/cc-pVDZ level in which the three-dimensional momentum vectors for all atoms were obtained. Propagation was performed for 500 fs with a time step of 0.5 fs. The total kinetic energy release (KER) can be obtained as the sum of the kinetic energies of the simulated molecular ions at this instant plus the remaining Coulomb potential energy considering the center-of-mass (COM) distance. The momentum vectors for three $C_6H_6^+$ cations are further obtained to generate the Dalitz plots and Newton diagrams of the corresponding three-body dissociation processes. In addition, the triple-ionization potential of benzene trimer was computed as the energy difference between the neutral $(C_6H_6)_3$ and the $C_6H_6^+ \cdot C_6H_6^+ \cdot C_6H_6^+$ trication states, which amounts to 35.9 eV at the CCSD(T)/cc-pVDZ level.

### Calculation of the sequential ionization cross-section

The absolute cross-section for sequential ionization (SI) leading to three fragment $C_6H_6^+$ ions was calculated by multiplying the partial ionization cross-section for the first collision partner with the probability of subsequently ionizing the second and third molecules. The latter is the relative size of its ionization cross-section compared with the total surface area of a sphere with a radius of the intermolecular distance:

$$\sigma_{SI}(E_0) = \sigma_{C_6H_6}^+(E_0) \times \frac{2 \cdot \sigma_{C_6H_6}^+(E_1)}{4\pi R^2} \times \frac{\sigma_{C_6H_6}^+(E_2)}{4\pi R^2} \quad (1)$$

Here $\sigma_{C_6H_6}^+(E_0) \approx \sigma_{C_6H_6}^+(E_1) \approx \sigma_{C_6H_6}^+(E_2) \approx 3.2 \, \text{Å}^2$ [39] is the partial ionization cross-section for the production of the intact $C_6H_6^+$ ion at the impact energy of $E_0 = 260$ eV, $E_1 = 250.8$ eV, and $E_2 = 241.6$ eV, which considers the minimum energy loss in the collision-induced ionization processes. $R \sim 5.15$ Å is the intermolecular COM distance considering the cyclic structure of the benzene trimer[17,18]. As a result, we obtain $\sigma_{SI} \approx 5.9 \times 10^{-4}$ Å².

### Calculation of the dSI+ICD cross-section

The cross-section of double sequential ionization plus ICD (dSI+ICD) can be calculated as:

$$\sigma_{dSI+ICD}(E_0) = \sigma_{C_6H_6}^+(E_0) \times \frac{2 \cdot \sigma_{IV}^+(E_1)}{4\pi R^2} \quad (2)$$

Here $\sigma_{IV}^+(E_1)$ is the inner-valence ionization cross-section. First, we determine a ratio of about 0.12 between the cross-sections for the inner-valence ionization (24−26 eV binding energy, giving rise to ICD) and for the outer-valence ionization (8−15 eV binding energy, leading to the intact $C_6H_6^+$ ion)[40]. Assuming this inner-shell vacancy decays completely by ICD, we determine the cross-section for dSI+ICD as: $\sigma_{dSI+ICD} \approx 7.4 \times 10^{-3}$ Å².

### Calculation of the dICD cross-section

We calculated the cross-sections of dICD in benzene trimers using two different methods. One is based on the yield ratio between two fragmentation pathways of the trimers, i.e., $C_6H_6^+ + (C_6H_6)_2^+$ and $C_6H_6^+ + C_6H_6^+ + C_6H_6^+$ channels, which were measured simultaneously in the experiment (Method 1). The other method is considering the theoretical asymptotic expressions for the decay width of dICD obtained in ref. 43 (Method 2). The dICD cross-section ($\sigma_{dICD}$) calculated with the experimental yields amounts to roughly $1.28 \times 10^{-2}$ Å², while we determine a possible lower limit for the $\sigma_{dICD}$ to be roughly $1.06 \times 10^{-3}$ Å² using the other theoretical method. Both indicate that dICD can play an important role in the three-body Coulomb explosion of benzene trimers ($C_6H_6^+ + C_6H_6^+ + C_6H_6^+$).

Method 1: In our experiments, the $C_6H_6^+ + (C_6H_6)_2^+$ channel can be attributed to the single ICD and SI of the trimer. The $C_6H_6^+ + C_6H_6^+ + C_6H_6^+$ channel is found to be caused by the dSI+ICD and the dICD processes. We notice that the ETMD process is not included in this channel due to both the larger intermolecular distances in benzene trimer and the competing Coulomb explosion processes of the initial dicationic state of the molecule. The yield ratio between $C_6H_6^+ + C_6H_6^+ + C_6H_6^+$ and $C_6H_6^+ + (C_6H_6)_2^+$ is determined to be 1 : 60, which can be described as:

$$\frac{\sigma_{dSI+ICD} + \sigma_{dICD}}{\sigma_{ICD} + \sigma_{SI}} = \frac{Y_{3 \times (C_6H_6^+)}}{Y_{C_6H_6^+ \cdot (C_6H_6)_2^+}} = \frac{1}{60} \quad (3)$$

where the cross-section $\sigma_{dSI+ICD}$ has been determined to be $7.4 \times 10^{-3}$ Å², and $\sigma_{ICD}$ and $\sigma_{SI}$ are the cross-sections of single ICD and SI leading to $C_6H_6^+ \cdot (C_6H_6)_2^+$ state of the trimer, respectively, which can be described as:

$$\sigma_{ICD} = 3 \times \sigma_{IV}^+(E_0) \quad (4)$$

$$\sigma_{SI} = \sigma_{C_6H_6}^+(E_0) \times \frac{2 \cdot \sigma_{C_6H_6}^+(E_1)}{4\pi R^2} \quad (5)$$

Where $\sigma_{IV}^+(E_0) = 0.384$ Å² is the inner-valence ionization cross-section[40], and $\sigma_{C_6H_6}^+(E_0) \approx \sigma_{C_6H_6}^+(E_1) = 3.2$ Å² are the partial ionization cross-sections for the production of intact $C_6H_6^+$ ion at the impact energies of $E_0 = 260$ eV and $E_1 = 250.8$ eV, respectively[39]. Thus, we can

determine $\sigma_{ICD} = 1.152\,\text{Å}^2$, $\sigma_{SI} = 6.15 \times 10^{-2}\,\text{Å}^2$. As a result, we obtain $\sigma_{dICD} = 1.28 \times 10^{-2}\,\text{Å}^2$.

Method 2: After the initial ionization of the deep-lying $C2s^{-1}$ states with binding energies above 35.9 eV, the system can relax through intermolecular decay (ICD and dICD) or intramolecular Auger processes. We consider a ratio of 5:1 between intermolecular decay and intramolecular Auger processes, which was obtained by comparing the experimental yields of $C_6H_6{}^+ + C_6H_6{}^+$ (intermolecular decay), $C_6H_6{}^{++}$ (Auger), and $C_6H_2{}^+/C_6H_3{}^+$ (inner-valence ionization of benzene) channels, as discussed in ref. [16]. Thus, the cross-section for the intermolecular decay processes ($\sigma_{ICD+dICD}$) can be determined as:

$$\sigma_{ICD+dICD} = \sigma_{IV}^+(E_0) \times \frac{5}{6} \qquad (6)$$

Where $\sigma_{IV}^+(E_0)$ is the inner-shell ionization section, which can be determined to be roughly 1% of $\sigma_{C_6H_6}^+(E_0)$, i.e., $3.2 \times 10^{-2}\,\text{Å}^2$, leading to a $C_2H^+$ fragment in the ionization of benzene monomer[50]. During the intermolecular decay processes, an outer-valence electron fills the inner-shell hole, followed by the transfer of a virtual photon ($E_p = 26.8$ eV)[62] to the neighboring molecules, which can cause the ionization of one or two benzene molecules, i.e., ICD or dICD, respectively. According to the asymptotic expressions for the decay width of dICD in ref. [43], their ratio can be provided by the double- to single-ionization cross-section ratio at the respective excess energy ($E_p$):

$$\frac{\sigma_{dICD}}{\sigma_{ICD}} = \frac{\sigma_{C_6H_6{}^+ \cdot C_6H_6{}^+}(E_p)}{\sigma_{(C_6H_6 \cdot C_6H_6)^+}(E_p)} \qquad (7)$$

For the photon energy of $E_p = 26.8$ eV, the $C_6H_6{}^+ \cdot C_6H_6{}^+$ state can be reached via ICD, knock-out (KO), and shake-off (SO) mechanisms. Here, we can estimate the cross-sections for ICD and KO, while the cross-section for SO is not obtained due to the lack of the cross-section of double ionization by a single photon for a benzene dimer or a similar dimer. According to experiments in ref. [42], the SO-type of ionization mechanism can be significant for the dICD process of a trimer system. Therefore, we can only estimate a possible lower limit for the $\sigma_{dICD}$ with this method. The ratio can be further described as:

$$\frac{\sigma_{C_6H_6{}^+ \cdot C_6H_6{}^+}(E_p)}{\sigma_{(C_6H_6 \cdot C_6H_6)^+}(E_p)} = \frac{(\sigma_{ICD} + \sigma_{KO})(E_p)}{\sigma_{(C_6H_6 \cdot C_6H_6)^+}(E_p)} \qquad (8)$$

Where the ICD cross-section can be determined as $\sigma_{ICD} = 2 \times \sigma_{IV}^+(E_p) = 2.1 \times 10^{-2}\,\text{Å}^2$ using the inner-valence ionization cross-section at photon energy of $E_p = 26.8$ eV[50]. The cross-section of KO can be calculated as:

$$\sigma_{KO}(E_p) = 2 \times \sigma_{C_6H_6}^+(E_p) \times \frac{\sigma_{C_6H_6}^+(E_2)}{4\pi R^2} \qquad (9)$$

where $\sigma_{C_6H_6}^+(E_p)$ amounts to 0.3 Å$^2$, corresponding to the single photon ionization cross-section at $E_p = 26.8$ eV[50], which emits an electron with energy of $E_2 = 26.8 - 9.2 = 17.6$ eV. This electron can cause the ionization of the third molecule with a probability determined as $\frac{\sigma_{C_6H_6}^+(E_2)}{4\pi R^2}$, where $\sigma_{C_6H_6}^+(E_2) = 2.06$ Å$^2$ is the ionization cross-section by electron-impact at 17.6 eV[39]. As a result, we obtain $\sigma_{KO}(E_p) = 3.72 \times 10^{-3}\,\text{Å}^2$. We further determine the single-ionization cross-section of benzene dimer as: $\sigma_{(C_6H_6 \cdot C_6H_6)^+}(E_p) = 2 \times \sigma_{C_6H_6}^+(E_p) = 0.6$ Å$^2$. This results in a lower limit of $\sigma_{dICD} \sim 1.06 \times 10^{-3}\,\text{Å}^2$.

In addition, when the molecule is doubly ionized by the energy released, the other dICD final state $(C_6H_6{}^+ \cdot C_6H_6{}^{++} \cdot C_6H_6)$ is formed, but that requires higher threshold energy of about 39.4 eV to be accessible.

## Data availability
All data supporting the findings of this study are available within the paper and its supplementary information files. Source data are provided with this paper.

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

## Acknowledgements

This work was supported by the National Natural Science Foundation of China (NSFC) under Grants No. 11974272 (J.Z., X.X., S.J., and X.R.), No. 11774281 (X.R.), and No. 12074143 (X.Y., S.L., and X.Z.). X.R. is grateful for support from the Open Fund of the State Key Laboratory of High Field Laser Physics (Shanghai Institute of Optics and Fine Mechanics).

## Author contributions

X.R. and D.D. conceived and supervised the project. J.Z., X.X., S.J., Y.Z., X.H., and X.R. performed the experiments using an electron source. X.Y., S.L., X.Z., L.H., C.W., and D.D. performed the experiments using a strong-field laser. J.Z., S.L., X.X., S.J., X.H., and X.R. carried out the molecular dynamics simulations. All authors discussed the results and commented on the manuscript.

## Competing interests

The authors declare no competing interests.
