## [Peer Review File · Nature Communications]

Triple ionization and fragmentation of benzene trimers following ultrafast intermolecular Coulombic decay

J. Zhou et al.,

Peer review of a draft submitted to: Nature Communications, (May 2022)

The authors report on the triple ionization of benzene trimers while imaging the momenta of three $C_6H_6^+$ (B_z^+) fragments in coincidence upon ionization with short electron and ultrashort laser pulses in two separated experiments. The interpretation of the measurements is guided by ab initio molecular dynamics (AMID) simulations.

The authors aim to identify the most likely conformer that was ionized. Eight conformers are identified by the AMID calculations as possible candidates. Observables like the kinetic energy release and the relative fragmentation angles between the ions were used and depicted in Dalitz plots and Newton diagrams. There appears to be reasonable agreement between measurement and simulations on two possible geometries that have contributed to the ion signal. In terms of ionization mechanisms the authors are trying to prove that double Intermolecular Coulombic Decay (dICD) has taken place and plays a significant role in the fragmentation process.

This is interesting work with rare differential insight, which I can likely see being published. However, I am not convinced (yet) enough evidence is given that dICD really happened in the triple ionization of benzene trimers and that new insight is found, which would justify publication in Nat. Comm. At this point I have some questions and remarks that need to be answered before I can make a more informed assessment and decision.

Please address the following questions and comments listed below and accept my suggestions as attempts to improve the current draft:

- Without measuring the electron energies, like it was realized in ref. [40], it is unclear to me how the authors can identify if dICD has really taken place. As ionization mechanisms are at the heart of this work, which may warrant the publication in Nat. Comm., it is important to line out the energetics and possible competing decay pathways in more detail, and perhaps even visualize them in a sketch similar to Fig. 1 in Ref. [39], in order to convince the reader that dICD is actually possible in this experiment. This would still not constitute the proof that dICD took place, but it is an important prerequisite.
- The authors acknowledge that competing Auger decay can happen as well, stating: “**This means that the system can also relax via intramolecular Auger decay and forms a $C_6H_6^{++}$ -(C_6H_6)₂ state. Here, we consider a ratio of about 40% for the decaying by intermolecular processes, which is predicted by a recent calculation on hydrated biomolecules with carbon core vacancies [42].**” In the reference cited it was *estimated* that “**For a pyrimidine molecule solvated by only four water molecules, our calculations predict a remarkable ratio of 58% of carbon core vacancies decaying by intermolecular processes. This value grows from 0% to 24%, 41%, and 50% for solvation by 0, 1, 2, and 3 water molecules, respectively....**” How can this *prediction* be translated one-to-one to the C_6H_6 trimer? Again, I think it would be necessary to discuss energetics in more detail. Can dICD be distinguished from Auger decay in the actual measurement? What would be the necessary observables to do so?

- Fig. 1(d) shows the measured time-of-flight (TOF) correlation map between the first detected $C_6H_6^+$ ion and the sum of the second and third detected $C_6H_6^+$ cations. However, it seems to have a hard momentum conservation gate applied on it, artificially creating a sharp line (or stripe), which suggests the measurement of three correlated Bz^+ cations. Without seeing the raw spectrum (without conditions) it is hard to judge if the 3-body breakup channel is really happening during the experiment and just needs to be isolated, or if this line of correlated ions is made up of uncorrelated ions passing the finite momentum conservation gates. The latter may have happened due to a rather warm supersonic gas jet target and the broadening of the momentum sum due to the recoil momentum of the particle collision. It may be instructive to show and discuss the momentum sum in the three lab frame coordinates x , y , and z .
- In this regard the authors should also look into the ion momentum position in the jet direction of their detector as a function of the time of flight. This spectrum will tell us about the composition of the supersonic jet, i.e. inform us if larger clusters were present and may contribute to producing multiple cations per shot.
- It would also be interesting to see the branching ratios, i.e. the relative yield differences in producing other ionization and breakup channels like, e.g., $C_6H_6^+$, $C_6H_6^{++}$, $C_6H_6^+-(C_6H_6)_2^+$. Is there a possibility of hydrogen loss, and would the resolution allow for detecting that? Would hydrogen loss shift or broaden the islands in the Dalitz plots in Fig. 3 and perhaps contribute to the broadened measured contribution?
- I don't think I understand the calculated Newton plots 4(d-i). How do they compare to the measured one in panel (a) when the reference ion is the slow ion (i.e. the one with the smallest momentum)?

So, to sum up, at the moment the information presented in the draft makes it hard for me to believe some of the conclusions and claims. Additional evidence and explanations will be helpful.

Additional minor comments:

- “Our analysis on the absolute cross-sections indicates the SI is only a minor channel...” should better read “Our analysis on the calculated absolute cross-sections...”
- “The fragmentation dynamics of the three-body Coulomb explosion process of benzene trimers are shown in Fig. 6.” It should better read “The calculated fragmentation dynamics...”

Journal: Nat.Comm. **Manuscript #:** NCOMMS-22-15838

Title : Triple ionization and fragmentation of benzene trimers following ultrafast intermolecular Coulombic decay

Authors: Jiaqi Zhou, Xitao Yu, Sizuo Luo, Xiaorui Xue, Shaokui Jia, Xinyu Zhang, Yongtao Zhao, Xintai Hao, Lanhai He, Chuncheng Wang, Dajun Ding, and Xueguang Ren

Report of referee

The knowledge of multiple ionization and fragmentation of dimers, trimers etc of aromatic systems is important for several fields including the field of radiation damage. Here, it is by now accepted that a substantial portion of the damage following high energy radiation is caused by the electrons emitted after ionization, Auger decay and intermolecular Coulombic decay.

Clearly, investigating the various –still uncovered- possibilities of multiple ionization and fragmentation of the above mentioned systems by impinging electrons is highly relevant.

The authors study the triple ionization of benzene trimer after bombardment with electrons of energy less than that needed to ionize a carbon core electron. Consequently, they can safely concentrate on outer- and inner-valence ionization. They show that triply ionizing the trimer by triple sequential ionization is less efficient by an order of magnitude than by double sequential ionization followed by intermolecular Coulombic decay. This is indeed an important finding which to my opinion justifies publication in Nature Communication.

What I miss is a discussion of the direct production (not via Auger) of dications of the kind $C_6H_6^{++}$. C_6H_6 . C_6H_6 by the impinging electrons. I checked and found that there are dicationic states of benzene which have sufficiently large double ionization potentials (>35.9 eV) in order to undergo ETMD. In ETMD, one of the neutral benzenes transfers an electron to $C_6H_6^{++}$, making it become a monocation $C_6H_6^+$, and the gain in energy leads to the ionization of the other neutral benzene. The result is $C_6H_6^+$. $C_6H_6^+$. $C_6H_6^+$, i.e., exactly the same product as discussed in the manuscript.

The estimates the authors give for the cross sections to triply ionize the trimer by triple sequential ionization and by double sequential ionization followed by intermolecular Coulombic decay are satisfactory.

The authors also discuss the process of double intermolecular Coulombic decay. This process directly leads to $C_6H_6^+ \cdot C_6H_6^+ \cdot C_6H_6^+$. Interestingly, they argue that the respective cross sections is the largest of all three processes they discuss. Until now, there is little data on this process. It would be a real 'sensation' if double intermolecular Coulombic decay is the dominant process for triple ionization of trimers made of aromatic molecules. Unfortunately, I do not find the estimate for the respective cross section made by the authors to be satisfactory. I suggest that the authors have a closer look at their ref.41 where a *lower bound* for the respective cross section is given. There, contact is made between double intermolecular Coulombic decay and double ionization by a single photon. Maybe the authors can find literature on the cross section of double ionization by a single photon for a benzene dimer or a similar dimer. This can strengthen the argumentation given in the manuscript.

As I mentioned above, I think that the work is highly relevant once the issue of producing high lying $C_6H_6^{++} \cdot C_6H_6 \cdot C_6H_6$ is appropriately discussed and, in addition, the issue of double intermolecular Coulombic decay is clarified. Even if this interesting process is found not to be the dominant one, the work has a high value. I await a revised version.

Response to Referees

General comments by the authors: We thank the Reviewers for their constructive criticism and comments. We have revised the manuscript and substantiated our findings according to the points raised. Some essential amendments are:

1. Due to the dilute nature of the benzene trimer in the gas jet and the low cross-section of the triple-ionization processes, we added more experiments to improve the experimental statistics.
2. We added an additional competing decay mechanism (suggested by Reviewer #2), i.e. electron transfer mediated decay (ETMD), and elucidated the detailed energetics for all possible decay mechanisms including a new schematic figure (suggested by Reviewer #1).
3. We recalculated the dICD cross-section (σ_{dICD}) based on the measured yields of two fragmentation pathways of benzene trimer, i.e. $\text{C}_6\text{H}_6^+ + (\text{C}_6\text{H}_6)_2^+$ and $\text{C}_6\text{H}_6^+ + \text{C}_6\text{H}_6^+ + \text{C}_6\text{H}_6^+$ channels, as suggested by Reviewers #1 and #2. This results a σ_{dICD} of about $1.28 \times 10^{-2} \text{ \AA}^2$, which can indicate that dICD is an important decay mechanism contributing to the three-body Coulomb explosion of benzene trimer.
4. We modified the Dalitz plots and Newton diagrams in Figs. 3 and 4 of the revised manuscript (suggested by Reviewer #1) by setting randomly one of the three fragment ions as reference since they are indistinguishable.
5. We determined a possible upper limit of the ICD lifetime in the SI+ICD channel using a classical model, which amounts to around 239 fs.

Reviewer #1 (Comments for the Author):

Main comments:

The authors report on the triple ionization of benzene trimers while imaging the momenta of three C_6H_6^+ (Bz^+) fragments in coincidence upon ionization with short electron and ultrashort laser pulses in two separated experiments. The interpretation of the measurements is guided by ab initio molecular dynamics (AMID) simulations.

The authors aim to identify the most likely conformer that was ionized. Eight conformers are identified by the AMID calculations as possible candidates. Observables like the kinetic energy release and the relative fragmentation angles between the ions were used and depicted in Dalitz plots and Newton diagrams. There appears to be reasonable agreement between measurement and simulations on two possible geometries that have contributed to the ion signal. In terms of ionization mechanisms the authors are trying to prove that double Intermolecular Coulombic Decay (dICD) has taken place and plays a significant role in the fragmentation process.

This is interesting work with rare differential insight, which I can likely see being published. However, I am not convinced (yet) enough evidence is given that dICD

really happened in the triple ionization of benzene trimers and that new insight is found, which would justify publication in Nat. Comm. At this point I have some questions and remarks that need to be answered before I can make a more informed assessment and decision.

Please address the following questions and comments listed below and accept my suggestions as attempts to improve the current draft.

Our reply: *We thank the Reviewer for the very positive assessment of our manuscripts. We have considered carefully these questions and comments in the following point-to-point response.*

1) Without measuring the electron energies, like it was realized in ref. [40], it is unclear to me how the authors can identify if dICD has really taken place. As ionization mechanisms are at the heart of this work, which may warrant the publication in Nat. Comm., it is important to line out the energetics and possible competing decay pathways in more detail, and perhaps even visualize them in a sketch similar to Fig. 1 in Ref. [39], in order to convince the reader that dICD is actually possible in this experiment. This would still not constitute the proof that dICD took place, but it is an important prerequisite.

Our reply: *i) We agree with the Reviewer that measuring the electron energies can help to identify the ionization mechanisms, in particular for the systems involving simple atoms. For molecular complexes, the ionization spectra become complicated for both initial and final states of the decay process. As a result, the kinetic energy distributions of dICD electrons become less identifiable for the study of benzene trimers. Additionally, there are four outgoing electrons in the present experiment, i.e., scattered projectile, directly ionized electron and two dICD electrons. Measuring these electrons in coincidence with three fragment $C_6H_6^+$ cations is still a challenging task. In the revised manuscript, we have recalculated the cross sections of dICD in benzene trimers using two different methods. One is based on the yield ratio between two fragmentation pathways of benzene trimers, i.e., $C_6H_6^+ + (C_6H_6)_2^+$ and $C_6H_6^+ + C_6H_6^+ + C_6H_6^+$ channels, which were measured simultaneously in the experiment. The other method is considering the theoretical asymptotic expressions for the decay width of dICD obtained in ref. [43]. The dICD cross-section (σ_{dICD}) calculated with the experimental yields amounts to $1.28 \times 10^{-2} \text{ \AA}^2$, while we determine a possible lower limit for the σ_{dICD} to be roughly $1.06 \times 10^{-3} \text{ \AA}^2$ using the other theoretical method. Both indicate that dICD can play an important role in the $C_6H_6^+ + C_6H_6^+ + C_6H_6^+$ Coulomb explosion of benzene trimer. It should be noted that the shake-off mechanism (single photon double ionization of benzene dimer) is not included in the latter calculations, while according to experiments in ref. [42] this type of ionization mechanism can be significant for the dICD process of a trimer system. Details about the calculations of σ_{dICD} have been added in the Methods section of the revised manuscript, which are also given in the following points.*

ii) We have lined out the energetics and possible competing decay pathways, and made a new energy-related plot in Fig. 1 of the revised manuscript, which shows schematically the decay mechanisms, including sequential ionization (SI), sequential ionization plus intermolecular Coulomb decay (SI+ICD), double intermolecular Coulomb decay (dICD), and electron transfer mediated decay (ETMD).

2) The authors acknowledge that competing Auger decay can happen as well, stating: “This means that the system can also relax via intramolecular Auger decay and forms a $C_6H_6^{++}-(C_6H_6)_2$ state. Here, we consider a ratio of about 40% for the decaying by intermolecular processes, which is predicted by a recent calculation on hydrated biomolecules with carbon core vacancies [42].” In the reference cited it was estimated that “For a pyrimidine molecule solvated by only four water molecules, our calculations predict a remarkable ratio of 58% of carbon core vacancies decaying by intermolecular processes. This value grows from 0% to 24%, 41%, and 50% for solvation by 0, 1, 2, and 3 water molecules, respectively....” How can this prediction be translated one-to-one to the C_6H_6 trimer? Again, I think it would be necessary to discuss energetics in more detail. Can dICD be distinguished from Auger decay in the actual measurement? What would be the necessary observables to do so?

Our reply: i) We now consider a ratio of 5:1 between intermolecular decay and intramolecular Auger processes, which was obtained by comparing the experimental yields of $C_6H_6^+ + C_6H_6^+$ (intermolecular decay), $C_6H_6^{++}$ (Auger), and $C_6H_2^+/C_6H_3^+$ (inner-valence ionization of benzene) channels, as discussed in ref. [16]. We use this ratio to determine the σ_{dICD} by considering the asymptotic expressions for the decay width of dICD in ref. [43].

ii) We elucidated the detailed energetics for all possible decay processes in the Introduction section of the revised manuscript (page 3, paragraphs 1 and 2). The discussions for dICD channel are as follows: “As is illustrated in Fig. 1c, the dICD reaction in benzene trimer is initiated by electron-collision with the removal of a deep-lying carbon 2s ($C2s$) inner-valence electron with binding energy above 36.0 eV. Afterward the $C_6H_6^{+*}-(C2s^{-1})-(C_6H_6)_2$ ionic state undergoes the dICD process, i.e. an electron from outer-valence shell of $C_6H_6^+$ fills the $C2s^{-1}$ vacancy, and the energy released (>26.8 eV) is sufficient to doubly ionize two neighboring molecules. Three $C_6H_6^+$ cations that are formed in the decay process repel each other, leading to Coulomb explosion of the system. We notice that the energy released may also singly or doubly ionize one neighboring molecules, resulting in $C_6H_6^+ \cdot C_6H_6^+ \cdot C_6H_6$ and $C_6H_6^+ \cdot C_6H_6^{++} \cdot C_6H_6$ states, respectively. However, neither of them can contribute to the present results and the latter process requires a higher threshold energy of about 39.4 eV to be accessible;”

iii) The intramolecular Auger process of benzene trimer forms a $C_6H_6^{2+} \cdot (C_6H_6)_2$ dication, which may further fragment into ionic and neutral species. While the dICD process leads to three coincident $C_6H_6^+$ cations which allows us to distinguish the dICD

channel from the Auger process in the experiment.

3) Fig. 1(d) shows the measured time-of-flight (TOF) correlation map between the first detected $C_6H_6^+$ ion and the sum of the second and third detected $C_6H_6^+$ cations. However, it seems to have a hard momentum conservation gate applied on it, artificially creating a sharp line (or stripe), which suggests the measurement of three correlated Bz^+ cations. Without seeing the raw spectrum (without conditions) it is hard to judge if the 3-body breakup channel is really happening during the experiment and just needs to be isolated, or if this line of correlated ions is made up of uncorrelated ions passing the finite momentum conservation gates. The latter may happen due to a rather warm supersonic gas jet target and the broadening of the momentum sum due to the recoil momentum of the particle collision. It may be instructive to show and discuss the momentum sum in the three lab frame coordinates x , y , and z .

Our reply: *The raw TOF correlation map without momentum conservation gates is now presented in Fig. 1e of the revised manuscript. The complete three-body Coulomb explosion channel ($C_6H_6^+ + C_6H_6^+ + C_6H_6^+$) is clearly observable as a sharp diagonal line in the time correlation map, which is well resolved from the $C_6H_6^+ + C_6H_6^+ + {}^{13}CC_5H_6^+$ and $C_6H_6^+ + C_6H_6^+ + C_6H_5^+ + H$ (H-loss) channels. It is noted that during the preparation of this reply we have performed more experiments to improve the experimental statistics. The momentum sum spectra in the three lab frame coordinates x , y , and z are presented in the following Figure 1. The peak widths (FWHM) of $\Delta p_{xsum} \sim \pm 6.0$ a.u. (Figure 1(a)), $\Delta p_{ysum} \sim \pm 7.0$ a.u. (Figure 1(b)), and $\Delta p_{zsum} \sim \pm 1.8$ a.u. (Figure 1(c)) are obtained, which are mainly caused by the temperature of the target and the recoil momentum of the electron collision. In the following Figure 2, we observe strong correlation features between two of the three $C_6H_6^+$ cations for the momenta ($p_{ion1} \sim p_{ion2} \sim 160$ a.u. in Figure 2 (a)) and the kinetic energies ($KE_{ion1} \sim KE_{ion2} \sim 2.5$ eV in Figure 2 (b)), which indicate a direct three-body fragmentation mechanism. Compared to the high momenta of the fragment $C_6H_6^+$ ions (~ 160 a.u.), the influence due to the temperature of the target and the recoil momentum of the electron collision can be negligible.*

Figure 1. The momentum sum of three $C_6H_6^+$ cations in the x -, y -, and z -directions of the lab coordinate.

Figure 2. The momentum (a) and kinetic energy (b) correlation of between two $C_6H_6^+$ ions of the three-body fragmentation channel.

4) In this regard the authors should also look into the ion momentum position in the jet direction of their detector as a function of the time of flight. This spectrum will tell us about the composition of the supersonic jet, i.e. inform us if larger clusters were present and may contribute to producing multiple cations per shot.

Our reply: We now present the y -coordinate (jet direction) position vs TOF 2D map in the following Figure 3(a), where the $C_6H_6^+$, $(C_6H_6)_2^+$ and $(C_6H_6)_3^+$ cations are identified in the experiment. The cations originating from pure ionization will exhibit a narrow peak in the TOF spectrum, which is the case for $(C_6H_6)_2^+$ in the pink area of Figure 3 (b), while the dissociative ionization of larger clusters will contribute as a broad TOF distribution, as shown the purple area in Figure 3 (b). For $(C_6H_6)_3^+$ in Figure 3 (c), there is no narrow peak, indicating that the ionization of benzene trimers is unstable. The observed $(C_6H_6)_3^+$ cations are mainly from the dissociative ionization of larger clusters (such as tetramer and larger clusters). The generation of $C_6H_6^+ + C_6H_6^+ + C_6H_6^+$ channel from triple ionization of larger clusters $(C_6H_6)_{n>3}$ requires evaporation of additional neutral C_6H_6 , which will appear as a broad pattern in the TOF correlation map like the H-loss channel shown in Fig. 1e of the revised manuscript. In the present experiment, we obtain a sharp correlation line for the $C_6H_6^+ + C_6H_6^+ + C_6H_6^+$ channel, which can exclude the contribution from larger clusters.

Figure 3. Y-coordinate (jet direction) position vs ion TOF map

5) It would also be interesting to see the branching ratios, i.e. the relative yield differences in producing other ionization and breakup channels like, e.g., $C_6H_6^+$, $C_6H_6^{++}$, $C_6H_6^+-(C_6H_6)_2^+$. Is there a possibility of hydrogen loss, and would the resolution allow for detecting that? Would hydrogen loss shift or broaden the islands in the Dalitz plots in Fig. 3 and perhaps contribute to the broadened measured contribution?

Our reply: We thank the Reviewer for the comments. i) We have analyzed the yields of $C_6H_6^+$, $C_6H_6^{++}$, $C_6H_6^+(C_6H_6)_2^+$, and $C_6H_6^+C_6H_6^+C_6H_6^+$ channels, and their ratios are determined to be 100 : 3.80 : 0.003 : 0.00005. We now consider the two fragmentation channels of benzene trimer, i.e. $C_6H_6^+(C_6H_6)_2^+$, and $C_6H_6^+C_6H_6^+C_6H_6^+$, to calculate the σ_{dICD} . The double ionization of benzene trimer leading to $C_6H_6^+(C_6H_6)_2^+$ channel can be attributed to the single ICD and sequential ionization (SI) of the trimer. The $C_6H_6^+C_6H_6^+C_6H_6^+$ channel is found to be caused by the SI+ICD and the dICD processes. We notice that the ETMD process is not included in this channel mainly because of the competing Coulomb explosion processes of the molecule in the initial dicationic state, in particular for the processes involving proton emission [50]. The yield ratio between $C_6H_6^+C_6H_6^+C_6H_6^+$ and $C_6H_6^+(C_6H_6)_2^+$ is determined to be 1 : 60, which can be described as:

$$\frac{\sigma_{SI+ICD} + \sigma_{dICD}}{\sigma_{ICD} + \sigma_{SI}} = \frac{Y_{3 \times C_6H_6^+}}{Y_{C_6H_6^+ \cdot (C_6H_6)_2^+}} = \frac{1}{60}$$

where $\sigma_{SI+ICD} = 7.4 \times 10^{-3} \text{ \AA}^2$ has been given in the Method section of the revised manuscript. σ_{ICD} and σ_{SI} are the cross sections of single ICD and SI leading to $C_6H_6^+ + (C_6H_6)_2^+$, respectively, which can be described as:

$$\sigma_{ICD} = 3 * \sigma_{IV}^+(E_0)$$

$$\sigma_{SI} = \sigma_{C_6H_6^+}(E_0) * \frac{2 * \sigma_{C_6H_6^+}(E_1)}{4\pi R^2}$$

Where $\sigma_{IV}^+(E_0) = 0.384 \text{ \AA}^2$ is the inner-valence ionization cross-section [40], and

$\sigma_{C_6H_6^+}(E_0) \approx \sigma_{C_6H_6^+}(E_1) = 3.2 \text{ \AA}^2$ is the partial ionization cross section for the production of intact $C_6H_6^+$ ion at the impact energy of 260 and 250.8 eV, respectively [39]. $R \sim 5.15 \text{ \AA}$ is the intermolecular distance. Thus, we can determine $\sigma_{ICD} = 1.152 \text{ \AA}^2$, and $\sigma_{SI} = 6.15 \times 10^{-2} \text{ \AA}^2$. As a result, we obtain $\sigma_{dICD} = 1.28 \times 10^{-2} \text{ \AA}^2$.

ii) We present the raw TOF correlation map in Fig. 1e of the revised manuscript. It can be seen from this 2D map that the complete three-body fragmentation channel ($C_6H_6^+ + C_6H_6^+ + C_6H_6^+$) is well resolved from the H-loss channel. This means that the contribution from the H-loss channel can be negligible in the present experiment.

6) I don't think I understand the calculated Newton plots 4(d-i). How do they compare to the measured one in panel (a) when the reference ion is the slow ion (i.e. the one with the smallest momentum)?

Our reply: We thank the Reviewer for the comments. We have modified the Dalitz plots and Newton diagrams in Figs. 3 and 4 of the revised manuscript by setting randomly one of the three fragment ions as reference since they are indistinguishable. In the Supplementary Fig. 2, we present the Newton diagram with a momentum range of ± 8 (arb. units) to visualize generally the three-body dissociation dynamics, in particular for the results with the slow ion as a reference. The diagrams for PDT, PDS, ST, S, T1, and T2 conformers show additional structures at larger momenta (> 3 arb. units). These structures are far from the range of the experimental pattern, and the calculations with C-trimer and the PD conformers agree well with the experimental results.

7) "Our analysis on the absolute cross-sections indicates the SI is only a minor channel..." should better read "Our analysis on the calculated absolute cross-sections..."

Our reply: This sentence has been revised accordingly: "Our analysis on the calculated absolute cross-sections indicates the SI is only a minor channel..."

8) "The fragmentation dynamics of the three-body Coulomb explosion process of benzene trimers are shown in Fig. 6." It should better read "The calculated fragmentation dynamics..."

Our reply: This sentence has been revised accordingly: "The calculated fragmentation dynamics of the three-body Coulomb explosion process of benzene trimers are shown in Fig. 6."

Reviewer #2 (Comments for the Author):

Main comments:

The knowledge of multiple ionization and fragmentation of dimers, trimers etc of aromatic systems is important for several fields including the field of radiation damage.

Here, it is by now accepted that a substantial portion of the damage following high energy radiation is caused by the electrons emitted after ionization, Auger decay and intermolecular Coulombic decay.

Clearly, investigating the various –still uncovered- possibilities of multiple ionization and fragmentation of the above mentioned systems by impinging electrons is highly relevant.

The authors study the triple ionization of benzene trimer after bombardment with electrons of energy less than that needed to ionize a carbon core electron. Consequently, they can safely concentrate on outer- and inner-valence ionization. They show that triply ionizing the trimer by triple sequential ionization is less efficient by an order of magnitude than by double sequential ionization followed by intermolecular Coulombic decay. This is indeed an important finding which to my opinion justifies publication in Nature Communication.

Our reply: *We thank the Reviewer for this very positive assessment of our manuscripts.*

1) What I miss is a discussion of the direct production (not via Auger) of dications of the kind $C_6H_6^{++}$. C_6H_6 . C_6H_6 by the impinging electrons. I checked and found that there are dicationic states of benzene which have sufficiently large double ionization potentials (>35.9 eV) in order to undergo ETMD. In ETMD, one of the neutral benzenes transfers an electron to $C_6H_6^{++}$, making it become a monocation $C_6H_6^+$, and the gain in energy leads to the ionization of the other neutral benzene. The result is $C_6H_6^+$. $C_6H_6^+$. $C_6H_6^+$, i.e., exactly the same product as discussed in the manuscript.

Our reply: *We now consider the ETMD process as a possible decay pathway leading to three $C_6H_6^+$ fragment ions, which is schematically shown in Fig. 1d of the revised manuscript and discussed in the Introduction section (Pages 3-4, Lines 77-88): “(iv) Electron transfer mediated decay (ETMD) [44-48] in which the outer-valence and inner-valence electrons are stripped from one molecule of the trimer. An electron from the neighboring molecules is transferred to fill the initial inner-valence vacancy and the energy released causes the ionization of the other neutral benzene (see in Fig.1d).*

There are three candidate states, i.e. $2e_{1u}^{-1}1e_{1g}^{-1}$, $2e_{1u}^{-1}3e_{2g}^{-1}$, and $2a_{1g}^{-1}1a_{2u}^{-1}$ lying above 36.0 eV binding energy [49], which are energetically accessible for the ETMD channel. It should be noted that since the decay width decreases exponentially with the intermolecular distance (R), the larger distance of $R \sim 5.15$ Å in benzene trimer can lead to a lower ETMD efficiency in comparison with that in the noble gas clusters [44-46]. Moreover, the molecule in these dicationic states may relax undergoing ultrafast Coulomb explosion prior to the electron transfer, in particular for the processes involving proton emission [50], which can suppress the occurrence of ETMD.”

2) The estimates the authors give for the cross sections to triply ionize the trimer by triple sequential ionization and by double sequential ionization followed by intermolecular Coulombic decay are satisfactory.

Our reply: *We thank the Reviewer for the positive assessment of our results.*

3) The authors also discuss the process of double intermolecular Coulombic decay. This process directly leads to $C_6H_6^+$. $C_6H_6^+$. $C_6H_6^+$. Interestingly, they argue that the respective cross sections is the largest of all three processes they discuss. Until now, there is little data on this process. It would be a real ‘sensation’ if double intermolecular Coulombic decay is the dominant process for triple ionization of trimers made of aromatic molecules. Unfortunately, I do not find the estimate for the respective cross section made by the authors to be satisfactory. I suggest that the authors have a closer look at their ref.41 where a lower bound for the respective cross section is given. There, contact is made between double intermolecular Coulombic decay and double ionization by a single photon. May be the authors can find literature on the cross section of double ionization by a single photon for a benzene dimer or a similar dimer. This can strengthen the argumentation given in the manuscript.

Our reply: *We thank the Reviewer for the suggestions. We have looked carefully through the ref. [43] in the revised manuscript and recalculated the dICD cross section (σ_{dICD}) using two different methods, among which one is based on the yield ratio between two fragmentation pathways of the trimers, i.e. $C_6H_6^+ + (C_6H_6)_2^+$ and $C_6H_6^+ + C_6H_6^+ + C_6H_6^+$ channels, which were measured simultaneously in the experiment (**Method 1**). The other method is considering the theoretical asymptotic expressions for the decay width of dICD obtained in ref. [43] (**Method 2**). The σ_{dICD} calculated with the experimental yields is roughly $1.28 \times 10^{-2} \text{ \AA}^2$, while for the other theoretical method we determine a possible lower limit for the σ_{dICD} to be roughly $1.06 \times 10^{-3} \text{ \AA}^2$. We notice that the shake-off mechanism (single photon double ionization of Benzene dimer) is not included in the latter calculations, while according to experiments in ref. [42] this type of ionization mechanism can be significant for the dICD process of a trimer system. Details about the calculations of σ_{dICD} have been added in the Method section of the revised manuscript, which are also given as follows:*

a) Method 1: *The $C_6H_6^+ + (C_6H_6)_2^+$ channel can be attributed to single ICD and sequential ionization (SI) of the trimer. The $C_6H_6^+ + C_6H_6^+ + C_6H_6^+$ channel is found to be caused by SI+ICD and dICD processes. We notice that the ETMD process is not included in this channel mainly because of the competing Coulomb explosion processes of the molecule in the initial dicationic states mentioned above. The yield ratio between $C_6H_6^+ + C_6H_6^+ + C_6H_6^+$ and $C_6H_6^+ + (C_6H_6)_2^+$ is determined to be 1 : 60, which can be described as:*

$$\frac{\sigma_{SI+ICD} + \sigma_{dICD}}{\sigma_{ICD} + \sigma_{SI}} = \frac{Y_{3 \times C_6H_6^+}}{Y_{C_6H_6^+ \cdot (C_6H_6)_2^+}} = \frac{1}{60}$$

where the $\sigma_{SI+ICD} \sim 7.4 \times 10^{-3} \text{ \AA}^2$ has been given in the Method section of the revised manuscript. σ_{ICD} and σ_{SI} are the cross sections of single ICD and SI leading to $C_6H_6^+ \cdot (C_6H_6)_2^+$ state of the trimer, respectively, which can be described as:

$$\sigma_{ICD} = 3 \times \sigma_{IV}^+(E_0)$$

$$\sigma_{SI} = \sigma_{C_6H_6^+}(E_0) \times \frac{2 \times \sigma_{C_6H_6^+}(E_1)}{4\pi R^2}$$

Where $\sigma_{IV}^+(E_0) = 0.384 \text{ \AA}^2$ is the inner-valence ionization cross-section [40], and $\sigma_{C_6H_6^+}(E_0) \approx \sigma_{C_6H_6^+}(E_1) = 3.2 \text{ \AA}^2$ are the partial ionization cross section for the production of intact $C_6H_6^+$ ion at the impact energies of $E_0 = 260$ and $E_1 = 250.8 \text{ eV}$, respectively [39]. $R \sim 5.15 \text{ \AA}$ is the intermolecular distance. Thus, we can determine $\sigma_{ICD} = 1.152 \text{ \AA}^2$, and $\sigma_{SI} = 6.15 \times 10^{-2} \text{ \AA}^2$. As a result, we obtain $\sigma_{dICD} = 1.28 \times 10^{-2} \text{ \AA}^2$.

b) Method 2: After the initial ionization of the deep-lying $C2s^{-1}$ states with binding energies above 36.0 eV, the system can relax through intermolecular decay (ICD and dICD) or intramolecular Auger processes. We consider a ratio of 5:1 between intermolecular decay and intramolecular Auger processes, which was obtained by comparing the experimental yields of $C_6H_6^+ + C_6H_6^+$ (intermolecular decay), $C_6H_6^{++}$ (Auger), and $C_6H_2^+/C_6H_3^+$ (inner-valence ionization of benzene) channels, as discussed in ref. [16]. Thus, the cross section for the intermolecular decay processes ($\sigma_{ICD+dICD}$) can be determined as:

$$\sigma_{ICD+dICD} = \sigma_{IV}^+(E_0) \times \frac{5}{6}$$

where $\sigma_{IV}^+(E_0)$ is the inner-shell ionization section, which can be determined to be roughly 1% of $\sigma_{C_6H_6^+}$, i.e. $3.2 \times 10^{-2} \text{ \AA}^2$, leading to a C_2H^+ fragment in the ionization of benzene monomer [50]. During the intermolecular decay processes, an outer-valence electron fills the inner-shell hole followed by the transfer of a virtual photon ($E_p = 26.8 \text{ eV}$) [66] to the neighboring molecules, which can cause the ionization of one or two benzene molecules, i.e., ICD or dICD, respectively. According to the asymptotic expressions for the decay width of dICD in ref. [43], their ratio can be provided by the double- to single-ionization cross section ratio at the respective excess energy, i.e.:

$$\frac{\Gamma_{dICD}}{\Gamma_{ICD}} = \frac{\sigma_B^{++}}{\sigma_B^+}$$

Here, the ratio between dICD and ICD processes is determined by the double- to single-ionization cross sections of benzene dimer at the photon energy of $E_p = 26.8 \text{ eV}$:

$$\frac{\sigma_{dICD}}{\sigma_{ICD}} = \frac{\sigma_{C_6H_6^+ \cdot C_6H_6^+}(E_p)}{\sigma_{(C_6H_6 \cdot C_6H_6)^+}(E_p)}$$

For the photon energy of $E_p = 26.8 \text{ eV}$, the $C_6H_6^+ \cdot C_6H_6^+$ state can be reached via ICD, knock-out (KO), and shake-off (SO) mechanisms. Here we can estimate the cross sections for ICD and KO, while the cross section for SO is not obtained due to the lack of the cross-section of double ionization by a single photon for a benzene dimer or a similar dimer. Therefore, we can only estimate a lower limit for the σ_{dICD} with this method. Thus, the ratio can be further described as:

$$\frac{\sigma_{C_6H_6^+ \cdot C_6H_6^+}(E_p)}{\sigma_{(C_6H_6 \cdot C_6H_6)^+}(E_p)} = \frac{(\sigma_{ICD} + \sigma_{KO})(E_p)}{\sigma_{(C_6H_6 \cdot C_6H_6)^+}(E_p)}$$

Where the ICD cross section can be determined as $\sigma_{ICD}(E_p) = 2 \times \sigma_{IV}(E_p) = 2.1 \times 10^{-2} \text{ \AA}^2$ using the inner-valence ionization cross-section at photon energy of $E_p=26.8 \text{ eV}$ [50]. The cross-section of KO can be calculated as:

$$\sigma_{KO}(E_p) = 2 \times \sigma_{C_6H_6^+}(E_p) \times \frac{\sigma_{C_6H_6^+}(E_2)}{4\pi R^2}$$

where $\sigma_{C_6H_6^+}(E_p)$ amounts to 0.3 \AA^2 , corresponding to the single photon ionization cross-section at $E_p=26.8 \text{ eV}$ [50], which emits an electron with energy of $E_2 = 26.8 - 9.2 = 17.6 \text{ eV}$. This electron can cause the ionization of the third molecule with a probability determined as $\frac{\sigma_{C_6H_6^+}(E_2)}{4\pi R^2}$, where $R \sim 5.15 \text{ \AA}$ is the intermolecular distance and $\sigma_{C_6H_6^+}(E_2) \sim 2.06 \text{ \AA}^2$ is the ionization cross section by electron impact at $E_2 = 17.6 \text{ eV}$ [39]. As a result, we obtain $\sigma_{KO}(E_p) = 3.72 \times 10^{-3} \text{ \AA}^2$. We further determine the single-ionization cross section of benzene dimer as: $\sigma_{(C_6H_6 \cdot C_6H_6)^+}(E_p) = 2 \times \sigma_{C_6H_6^+}(E_p) = 0.6 \text{ \AA}^2$. This results a possible lower limit of $\sigma_{dICD} \sim 1.06 \times 10^{-3} \text{ \AA}^2$.

In addition, when the molecule is doubly ionized by the energy released, the other dICD final state ($C_6H_6^+ \cdot C_6H_6^{++} \cdot C_6H_6$) is formed, but that requires a higher threshold energy of about 39.4 eV to be accessible.

4. As I mentioned above, I think that the work is highly relevant once the issue of producing high lying $C_6H_6^{++} \cdot C_6H_6 \cdot C_6H_6$ is appropriately discussed and, in addition, the issue of double intermolecular Coulombic decay is clarified. Even if this interesting process is found not to be the dominant one, the work has a high value. I await a revised version.

Our reply: We thank the Reviewer for this very positive assessment of our manuscripts. We elucidate more details on the competing decay mechanisms, in particular for the ETMD process in the Introduction section of the revised manuscript (Page 3, Paragraphs 1 and 2). We have recalculated the dICD cross-section mentioned above, which can indicate that dICD is an important process contributing to the three-body Coulomb explosion of benzene trimer.

Triple ionization and fragmentation of benzene trimers following ultrafast intermolecular Coulombic decay

J. Zhou et al.,

Peer review of a draft resubmitted to: Nature Communications, (July 2022)

I have read the response letter of the authors as well as the new draft and the supplemental information. My questions were addressed, and my comments and suggestions were implemented. The article has nicely improved. I find the results and interpretations very convincing. Looking at the electron impact experiment, I can now believe in seeing the fingerprint of dICD, which makes this work a viable candidate for publication in Nat. Comm.

The last obstacle for me is the comparison with the laser experiment, which is in principle a great idea and tool. The authors sum up the gist on **page 7, line 185** with “The obtained KER, Dalitz and Newton spectra ... are in good agreements with the main features of the electron-initiated experiments. These results can support that the electron-collision induced dICD and (d)SI+ICD processes can take place in the fs time scale, leading to the fast and concerted breakup of benzene trimers.”. However, the text sounds like as if the authors imply here that the dICD and (d)SI+ICD processes are also dominant in the laser experiment. The presented spectra in Fig. 5 and the comparison of the KER distributions of the electron and laser experiment also seem to suggest so.

However, in my mind, I expect the ionization mechanism with multiple 800 nm photons at intensities of $6E14$ W/cm² to differ from an inner valence ionization with a single electron of 260 eV. I would champion mechanisms like sequential 3xSI or a 2xSI, followed by a recollision ionization. One could perhaps envision a SI + recollision process which emits an innershell electron that opens the door for high harmonic generation which then ionizes a neighboring molecule. However, I can't easily see how a dICD could take place. So in its current form, this narrative is somewhat puzzling for the reader.

The authors should elaborate a little more on the possible laser ionization mechanisms and the agreement of the electron impact and laser results. This will help the comparison of the measured KER distributions to estimate an upper lifetime limit for the dissociating trimer dication till ICD sets in the electron impact experiment. The latter is in principle very clever and is summarized on **page 7, line 195** as “As in this process, after the initial SI the two cations in benzene trimer...begin to repel each other which can cause the increasing of the intermolecular distances before the ICD process.”. Since ICD is R-dependent, it would perhaps also explain why the shoulder is so much lower in yield than the main KER peak. However, with the open question on the dominating ionization mechanisms in the laser experiment, this interpretation appears somewhat questionable at this point. I think this is an easy fix though, which only requires a couple of clarifying sentences in the narrative. I would not have to see the draft again.

Besides that I only have minor suggestions:

- **Entire draft:** I suggest to perhaps call the SI+ICD process dSI+ICD (the “d” standing for “double”) to better reflect that two sequential SI processes and one ICD take place producing a triple ionization.

- **Page 3, line 72:** along the same lines as the suggestion above, I recommend changing “...is sufficient to doubly ionize the neighboring two molecules...” to “is sufficient to each single ionize the neighboring two molecules...”.
- **Page 3, line 76:** perhaps write “However, neither of them can contribute to the present results, which demands three $C_6H_6^+$ ions to be detected in coincidence, and the latter process requires a higher threshold energy of about 39.4 eV to be accessible;”
- **Page 4, line 105:** please write “...the recoil momentum of the impinging electron during the collision.”
- **Page 5, line 117+130:**
Line 117: in order to avoid confusion, please do not refer to “(Fig. 2b)” here, as the elastically predicted KER of 8.4 eV does not reflect the theoretical result of the AIMD simulations displayed in this panel.
Line 130: accordingly, now write “...of the benzene dimer: the cyclic (C), the sandwich (S), T-shaped (T), and...”
- **Page 6, line 170:** this sentence seems incomplete. Did you perhaps intend to write: “While the calculations for other conformers are far from the range of the experimental pattern in particular for the ST, S, T1, and T2 conformers, as shown in Figs. 4f-i, respectively, there may be small contributions from PDS and PDT conformers.”? In general, throughout the draft it appears PDS and PDT conformers could contribute to a small degree. This sentence may leave the door open for that until future experiments bring more clarity.
- **Page 7, line 106:** I suggest writing: “These spectra nicely visualize the concerted fragmentation of...”.
- **Page 8, line 218:** please write “Our calculations on the ionization cross-section after electron impact indicate...”
- **Fig. 1(e), 3(a), 4(a), 5(c), 5(d), SI 1(a+b):** please state in the figure captions if the yield color code of these 2d spectra is linear or on a logarithmic scale.

Reviewer #2 (Remarks to the Author):

Second Report of Referee #2

In my first report I concluded: "As I mentioned above, I think that the work is highly relevant once the issue of producing high lying $C_6H_6^{++}$ C_6H_6 C_6H_6 is appropriately discussed and, in addition, the issue of double intermolecular Coulombic decay is clarified. Even if this interesting process is found not to be the dominant one, the work has a high value. I await a revised version."

The authors have responded in detail to all my comments and as far as I can judge also to those of Referee #1.

I am satisfied with the response of the authors and with the revisions made in the manuscript.

In particular, the authors now discuss the importance of double intermolecular Coulombic decay much more convincingly by applying two different methods. It is now clear that double intermolecular Coulombic decay is operative and relevant in the triple ionization and fragmentation of benzene trimers.

The argumentation by the authors concerning the relevance of ETMD in the triple ionization and fragmentation of benzene trimers after bombardment with electrons is very sound too. I wonder whether the authors can carry out similar measurements on other trimers where energetically lower dicationic states can undergo ETMD and therefore be more efficient than in benzene. But this is only a suggestion for the future and does not relate directly to the present work.

Recommendation: I recommend the paper for publication in Nature Communications

Reviewer #1 (Comments for the Author):

Main comments:

I have read the response letter of the authors as well as the new draft and the supplemental information. My questions were addressed, and my comments and suggestions were implemented. The article has nicely improved. I find the results and interpretations very convincing. Looking at the electron impact experiment, I can now believe in seeing the fingerprint of dICD, which makes this work a viable candidate for publication in Nat. Comm.

Our reply: *We thank the Reviewer for the very positive assessment of our manuscripts.*

1) The last obstacle for me is the comparison with the laser experiment, which is in principle a great idea and tool. The authors sum up the gist on page 7, line 185 with “The obtained KER, Dalitz and Newton spectra ... are in good agreements with the main features of the electron-initiated experiments. These results can support that the electron-collision induced dICD and (d)SI+ICD processes can take place in the fs time scale, leading to the fast and concerted breakup of benzene trimers.”. However, the text sounds like as if the authors imply here that the dICD and (d)SI+ICD processes are also dominant in the laser experiment. The presented spectra in Fig. 5 and the comparison of the KER distributions of the electron and laser experiment also seem to suggest so. However, in my mind, I expect the ionization mechanism with multiple 800 nm photons at intensities of 6×10^{14} W/cm² to differ from an inner valence ionization with a single electron of 260 eV. I would champion mechanisms like sequential 3xSI or a 2xSI, followed by a recollision ionization. One could perhaps envision a SI + recollision process which emits an innershell electron that opens the door for high harmonic generation which then ionizes a neighboring molecule. However, I can't easily see how a dICD could take place. So in its current form, this narrative is somewhat puzzling for the reader.

Our reply: *We added two Refs. [60, 61] and elucidated more details on the ionization mechanisms of benzene trimer in the laser experiment: “In this experiment, the repulsive $C_6H_6^{++} + C_6H_6^+ + C_6H_6^+$ tricationic state is mainly caused by sequentially ionizing three molecules of the trimer during the laser pulse (3xSI) [60]. While the SI with additional recollision mechanisms can be ruled out as playing an important role for the triple ionization of benzene trimer since the recollision processes of the photoelectron are effectively suppressed by the circularly polarized light used in the fs laser experiment [61].” (Page 7, Lines 188-193 of the revised manuscript)*

2) The authors should elaborate a little more on the possible laser ionization mechanisms and the agreement of the electron impact and laser results. This will help

the comparison of the measured KER distributions to estimate an upper lifetime limit for the dissociating trimer dication till ICD sets in the electron impact experiment. The latter is in principle very clever and is summarized on page 7, line 195 as “As in this process, after the initial SI the two cations in benzene trimer...begin to repel each other which can cause the increasing of the intermolecular distances before the ICD process.” Since ICD is R-dependent, it would perhaps also explain why the shoulder is so much lower in yield than the main KER peak. However, with the open question on the dominating ionization mechanisms in the laser experiment, this interpretation appears somewhat questionable at this point. I think this is an easy fix though, which only requires a couple of clarifying sentences in the narrative. I would not have to see the draft again.

Our reply: *We thank the Reviewer for the supportive comments. We have elaborated on the laser ionization mechanism and the comparison between the electron-initiated and laser experiments. We added the sentences: “In the laser experiment, three outer-valence electrons are rapidly stripped from each molecule of the benzene trimer (< 40 fs), and the nuclear dynamics during the 3×SI is nearly frozen. The subsequent Coulomb explosion of the trimer can thus lead to a KER distribution similar to the main peak obtained in the dICD and dSI+ICD processes.” (Page 7, Lines 198-202 of the revised manuscript)*

Page 7, Lines 203-204: We modify the sentence, it now reads “... can take place in the fs regime, which is comparable to the timescale of the 3×SI occurring in the laser experiment.”

We added the sentence in Page 7 and Lines 204-207 of the revised manuscript: “It must be stressed here that the ionization mechanisms leading to the Coulomb explosion of benzene trimers are different between the electron-initiated and laser experiments, a dominant peak at KER ~ 7.4 eV can be observed in both experiments due to the ultrafast nature of their initial ionization processes.”

The possible reason for the lower yield of the shoulder is discussed at Page 8 and Lines 215-217 of the revised manuscript: “Since the ICD rate strongly depends on the intermolecular distance ($1/R^6$) [62], which can cause the lower yield of the shoulder at KER ~ 6.5 eV.”

Minor suggestions:

1) Entire draft: I suggest to perhaps call the SI+ICD process dSI+ICD (the “d” standing for “double”) to better reflect that two sequential SI processes and one ICD take place producing a triple ionization.

Our reply: *“SI+ICD” has been replaced by “dSI+ICD” throughout the whole manuscript.*

2) Page 3, line 72: along the same lines as the suggestion above, I recommend changing "...is sufficient to doubly ionize the neighboring two molecules..." to "is sufficient to each single ionize the neighboring two molecules..."

Our reply: *This sentence has been revised accordingly: "... doubly ionize ..." is changed to "each single ionize ..." (Page 3, Line 80 of the revised manuscript)*

3) Page 3, line 76: perhaps write "However, neither of them can contribute to the present results, which demands three $C_6H_6^+$ ions to be detected in coincidence, and the latter process requires a higher threshold energy of about 39.4 eV to be accessible;"

Our reply: *This sentence has been revised accordingly: we added "..., which demands three $C_6H_6^+$ ions to be detected in coincidence, ..." in the sentence. (Page 4, Lines 84-85 of the revised manuscript)*

4) Page 4, line 105: please write "...the recoil momentum of the impinging electron during the collision."

Our reply: *This sentence has been revised accordingly: we added "... of the impinging electron ..." in the sentence. (Page 4, Line 110 of the revised manuscript)*

5) Page 5, line 117+130:

Line 117: in order to avoid confusion, please do not refer to "(Fig. 2b)" here, as the elastically predicted KER of 8.4 eV does not reflect the theoretical result of the AIMD simulations displayed in this panel.

Line 130: accordingly, now write "...of the benzene dimer: the cyclic (C), the sandwich (S), T-shaped (T), and..."

Our reply: *These two sentences have been revised accordingly. Page5, Lines 123-124: We removed "(Fig. 2b)" from the sentence, it now reads: "For a symmetric triangular or cyclic (C) structure of the trimer, which is...."*

Page 5, Line 136: We added "... the cyclic (C), ..." in the sentence.

6) Page 6, line 170: this sentence seems incomplete. Did you perhaps intend to write: "While the calculations for other conformers are far from the range of the experimental pattern in particular for the ST, S, T1, and T2 conformers, as shown in Figs. 4f-i, respectively, there may be small contributions from PDS and PDT conformers."? In general, throughout the draft it appears PDS and PDT conformers could contribute to a small degree. This sentence may leave the door open for that until future experiments bring more clarity.

Our reply: *This sentence has been revised accordingly: we added in the end of the*

*sentence: “..., there may be small contributions from PDS and PDT conformers.”
(Page 6, Line 179)*

7) Page 7, line 106: I suggest writing: “These spectra nicely visualize the concerted fragmentation of...”.

Our reply: *This sentence has been revised accordingly: “... reveal clearly ...” is replaced by “... nicely visualize ...” (Page 8, Line 226 of the revised manuscript)*

8) Page 8, line 218: please write “Our calculations on the ionization cross-section after electron impact indicate...”

Our reply: *This sentence has been revised accordingly: we added “... after electron impact ...” in the sentence. (Page 8, Line 239)*

9) Fig. 1(e), 3(a), 4(a), 5(c), 5(d), SI 1(a+b): please state in the figure captions if the yield color code of these 2d spectra is linear or on a logarithmic scale.

Our reply: *We state that the color bar is linear in all the figure captions mentioned above: “The counts intensity is color-coded on a linear scale.”*